# Molecular Phylogeny and Evolution of Amazon Parrots in the Greater Antilles

**DOI:** 10.3390/genes12040608

**Published:** 2021-04-20

**Authors:** Sofiia Kolchanova, Alexey Komissarov, Sergei Kliver, Anyi Mazo-Vargas, Yashira Afanador, Jafet Velez-Valentín, Ricardo Valentín de la Rosa, Stephanie Castro-Marquez, Israel Rivera-Colon, Audrey J. Majeske, Walter W. Wolfsberger, Taylor Hains, André Corvelo, Juan-Carlos Martinez-Cruzado, Travis C. Glenn, Orlando Robinson, Klaus-Peter Koepfli, Taras K. Oleksyk

**Affiliations:** 1Biology Department, University of Puerto Rico at Mayagüez, Mayagüez 00682, Puerto Rico; sofiia.kolchanova@gmail.com (S.K.); bioanyimima@gmail.com (A.M.-V.); yashirampr@gmail.com (Y.A.); stephaniecastro@oakland.edu (S.C.-M.); israelriveracolon@gmail.com (I.R.-C.); amajeske@oakland.edu (A.J.M.); wwolfsberger@oakland.edu (W.W.W.); juancarlos.martinez@upr.edu (J.-C.M.-C.); 2Theodosius Dobzhansky Center for Genome Bioinformatics, St. Petersburg State University, 199034 St. Petersburg, Russia; koepflik@si.edu; 3Applied Genomics Laboratory, SCAMT Institute, ITMO University, 191002 St. Petersburg, Russia; komissarov@scamt-itmo.ru; 4Institute of Molecular and Cellular Biology, Siberian Branch of the Russian Academy of Sciences, 664033 Novosibirsk, Russia; mahajrod@gmail.com; 5Conservation Program of the Puerto Rican Parrot, U.S. Fish and Wildlife Service, Rio Grande 00745, Puerto Rico; jafet_velez@fws.gov; 6The Recovery Program of the Puerto Rican Parrot at the Rio Abajo State Forest, Departamento de Recursos Naturales y Ambientales de Puerto Rico, Arecibo 00613, Puerto Rico; el.cotorro.electrico@gmail.com; 7Department of Biological Sciences, Oakland University, Rochester, MI 48307, USA; 8Department of Biology, Uzhhorod National University, 88000 Uzhhorod, Ukraine; 9Terra Wildlife Genomics, Washington, DC 20009, USA; hainst2111@gmail.com; 10Environmental Science and Policy, Johns Hopkins University, Washington, DC 20036, USA; 11New York Genome Center, New York, NY 10013, USA; acorvelo@nygenome.org; 12Department of Environmental Health, The University of Georgia, Athens, GA 30602, USA; travisg@uga.edu; 13Hope Zoo and Botanical Gardens, Kingston 6, Jamaica; orlandofrobinson@yahoo.com; 14Center for Species Survival, Smithsonian Conservation Biology Institute, National Zoological Park, Front Royal, VA 22630, USA

**Keywords:** *Amazona*, Caribbean, mitochondria, genomes, *A. leucocephala*, *A. agilis*, *A. collaria*, *A. ventralis*, *A. vittata*, *A. rhodocorytha*

## Abstract

Amazon parrots (*Amazona* spp.) colonized the islands of the Greater Antilles from the Central American mainland, but there has not been a consensus as to how and when this happened. Today, most of the five remaining island species are listed as endangered, threatened, or vulnerable as a consequence of human activity. We sequenced and annotated full mitochondrial genomes of all the extant Amazon parrot species from the Greater Antillean (*A. leucocephala* (Cuba), *A. agilis*, *A. collaria* (both from Jamaica), *A. ventralis* (Hispaniola), and *A. vittata* (Puerto Rico)), *A. albifrons* from mainland Central America, and *A. rhodocorytha* from the Atlantic Forest in Brazil. The assembled and annotated mitogenome maps provide information on sequence organization, variation, population diversity, and evolutionary history for the Caribbean species including the critically endangered *A. vittata*. Despite the larger number of available samples from the Puerto Rican Parrot Recovery Program, the sequence diversity of the *A. vittata* population in Puerto Rico was the lowest among all parrot species analyzed. Our data support the stepping-stone dispersal and speciation hypothesis that has started approximately 3.47 MYA when the ancestral population arrived from mainland Central America and led to diversification across the Greater Antilles, ultimately reaching the island of Puerto Rico 0.67 MYA. The results are presented and discussed in light of the geological history of the Caribbean and in the context of recent parrot evolution, island biogeography, and conservation. This analysis contributes to understating evolutionary history and empowers subsequent assessments of sequence variation and helps design future conservation efforts in the Caribbean.

## 1. Introduction

Uncovering the underlying mechanisms of speciation is perhaps the most significant quest of evolutionary biology, and islands have been an essential source of observational as well as experimental data ever since Darwin’s famous voyage on the HMS Beagle [1,2,3]. In effect, island species provide the most valuable model systems for fundamental studies on migration, diversification, and extinction [4]. Since many individual islands are young and contain relatively few phylogenetic groups, evolutionary adaptations and species proliferation may be more obvious and easier to study compared to continental biota with large population sizes and boundaries that are difficult to define [5]. Additionally, the geographic isolation of many islands allows the evolution of resident species to take its own course, usually independent of influence from other areas, resulting in endemic faunas and floras [2,6]. As a result, high levels of biodiversity are observed on islands around the world. While comprising just 3.5% of the Earth’s land area, they are home to 15–20% of all terrestrial species [4], a large proportion of which are endemic. Unfortunately, island biotas are also extremely vulnerable to perturbation and destruction by humans, which, in combination with their rich diversity, contributes significantly to contemporary mass extinctions [7]. In this study, we take a look at an example of island speciation in the parrots of the Caribbean.

*Psittaciformes* (parrots, parakeets, macaws, and cockatoos) are the most threatened avian order [8,9,10,11] with many of the species classified as Critically Endangered, Endangered, and Vulnerable by the International Union for Conservation of Nature [12]. The biggest threats affecting parrot species come from farming and animal grazing by the agricultural industry, as well as capturing of individuals either for local or international pet trade [13]. Recent studies show that populations of more than one-third of all the Neotropical species are on the decline [13,14]. In the Caribbean, the decline of some species has recently slowed with the implementation of aggressive habitat protection and vigorous environmental education campaigns. However, a number of populations, such as the Cayman Brac Parrot (*Amazona*
*leucocephala hesterna*) and the Puerto Rican parrot (*A. vittata*), though showing stable population levels, still face many substantial conservation challenges and remain at high risk of rapid extinction [14]. In particular, *A. vittata* parrots almost became extinct in 1975 when their number dropped to only 13 birds [15], and recently, became especially vulnerable after major hurricanes caused severe damage to their habitat. They have been listed as endangered by the US Fish and Wildlife Service (USFWS) since 1967, and as critically endangered by the IUCN Red Book since 1994 [12].

Past molecular and morphological studies have suggested that all species of Amazon parrots (*Amazona spp*.) that currently inhabit the Greater Antillean islands of Cuba, Jamaica, Hispaniola, and Puerto Rico dispersed eastward from an ancestral population in Central America (Figure 1), while those native to the Lesser Antilles are likely the result of two independent northward dispersal events originating from the coast of northern South America [16]. Two alternative migration-speciation scenarios involving overwater dispersal from Central America have been proposed for the Greater Antillean *Amazona* parrots (Appendix A) [15,17]. Bond [18], Lantermann [19], and Snyder et al. [15] suggested a close relationship between *A. agilis* and *A. vittata* based on plumage characteristics (red forehead patch). This would imply direct colonization from Jamaica to Puerto Rico or a stepping-stone path with intermediate populations going extinct later on. The red forehead patch, however, is most likely a result of convergent evolution. Blue primary coverts are characteristic of *A. collaria, A. leucocephala*, *A. ventralis*, and *A. vittata*, and may be a derived plumage characteristic in the Greater Antillean *Amazona*, while red primary coverts in *A. agilis* could suggest that it is a separate lineage [20]. A close relationship of *A. vittata* to *A. ventralis*, and not *A. agilis*, makes more sense geographically and can be supported by plumage similarities other than the red forehead patch [15,20]. Lack’s view was that *A. ventralis* and *A. vittata* are sister lineages sharing a common ancestor with *A. leucocephala* [17] (Appendix A). This disagreement could be resolved using molecular data by inferring phylogenetic relationships among extant species, and the divergence between sequences could help determine the timing of speciation events along the dispersal routes. Only a couple of studies have previously attempted to reconstruct the evolutionary history of this avian clade, but their efforts were limited by the state of the sequencing technology, allowing only partial reconstruction of relatively short genomic and partial mitochondrial sequences [16,20].

Understanding the speciation and past evolutionary histories of Caribbean Amazon parrots is an important component in designing scientifically justified conservation strategies that would help mitigate current threats of extinction [21]. Detailed genetic information on species variability will help develop high-resolution molecular techniques to be used for uncovering critical information for the preservation of diversity and viability of parrot populations, such as species identity, degree of hybridization, genetic diversity, demographic history, and effective population size of living organisms [22]. In the last decade, the rapid development of NGS (Next-Generation Sequencing) technologies has made it possible to identify and interpret differences between closely related genomes in their evolutionary context [23,24].

In the current study, we took advantage of the available tools in the rapidly developing fields of NGS and bioinformatics to look at the patterns of speciation and recent evolutionary history of six Caribbean Amazon parrots. We sequenced, assembled, and annotated full mitochondrial genomes from the Greater Antillean Amazons, starting with the mitogenome of *Amazona vittata* [25,26]. Using the new data, we addressed long-standing issues about the phylogenetic relationships within the clade, evaluated two alternative speciation hypotheses for parrots in the Greater Antillean islands (Appendix A) and evaluated genetic diversity in several species by comparing multiple individual mitogenomes. The data we present can be used in future conservation studies to determine identity, population genetics, and to help drive ongoing efforts to preserve the wild and captive populations of these highly charismatic endemic species of birds.

**Figure 1 genes-12-00608-f001:**
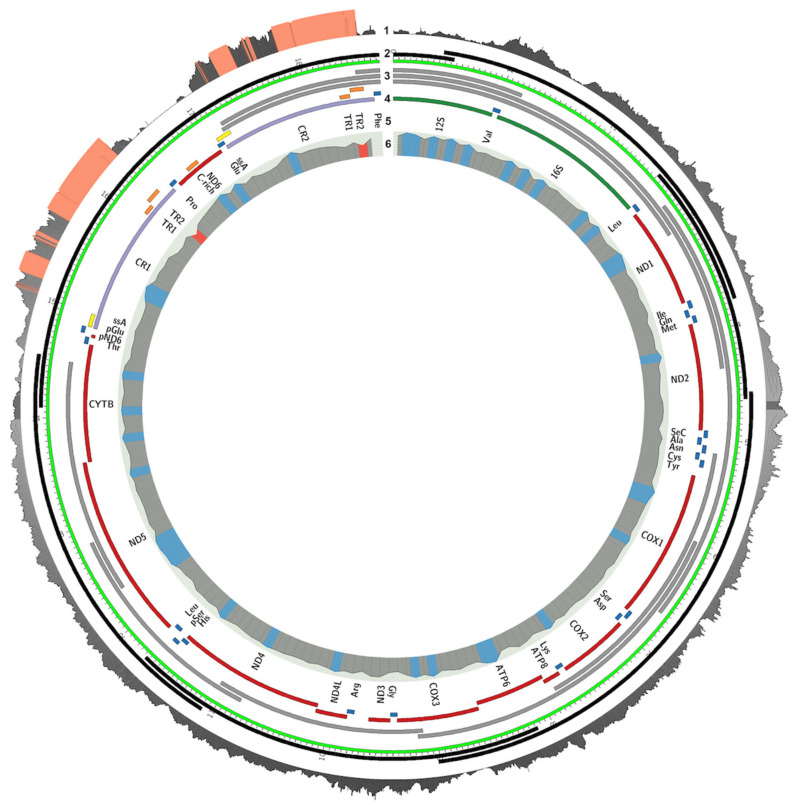
The annotated assembly of the Puerto Rican parrot (*Amazona vittata*) mitogenome. The concentric tracks show (from the outside in): (1) Coverage by mapped Illumina paired-end reads. The orange color indicates a coverage depth of >140 reads/bp; (2) Mapped PacBio reads (black); the green ruler denotes the chromosome and its coordinates; (3) Sanger sequence reads and PCR products used for distance validation; (4) Feature annotations: (red) protein-coding genes, (green) rRNA genes, (blue) tRNA genes, (purple) control region, (orange) tandem repeats, (yellow) ssA; (5) gene names; (6) GC content (red if <30%, blue if >50%). The figure was generated using Circos [27].

## 2. Materials and Methods

### 2.1. Sample Collection and DNA Extraction

This study was reviewed by, and all the samples were obtained with, the approval of the Institutional Animal Care and Use Committee of the University of Puerto Rico following the guidance for the Endangered Species Act (IACUC# 201109.1). In Jamaica, permits were issued by the Jamaican National Environment and Planning Agency (NEPA). In the USA and Puerto Rico, the required collection and export permits were issued by the US government under the Endangered Species Act (ESA), the Convention on International Trade in Endangered Species of Wild Fauna and Flora (CITES), and by the Animal and Plant Health Inspection Service (APHIS) before any work commenced.

Sources, origins, and numbers of the samples used in this study are listed in Appendix A. *Amazona vittata* (n = 10) and *A. ventralis* (n = 2) blood samples were obtained from the US Fish and Wildlife Service (USFWS) aviary at El Yunque National Forest at Luquillo, Puerto Rico, as discarded blood leftovers from routine veterinary procedures. Samples of the two Jamaican parrot species (*A. agilis* (n = 3) and *A. collaria* (n = 7)) were also obtained from material discarded after routine veterinary procedures from the rescued birds recovering at the Hope Gardens and Zoo in Kingston, Jamaica. The sample of *A. leucocephala* (n = 1) originated from the Frozen Zoo^®^ collection at the San Diego Zoo Institute for Conservation Research. The white-fronted amazon (*A. albifrons*, Yucatan peninsula, Mexico), and the red-browed amazon (*A. rhodocorytha*, Brazil) samples consisted of plucked feathers mailed by the owners of pet birds in private collections to the University of Puerto Rico at the personal request of the aviculturists from the USFWS Puerto Rican Parrot Recovery Program. For all samples, genomic DNA was extracted using the DNeasy Blood and Tissue Kit (Qiagen Inc., Valencia, CA, USA) following the manufacturer’s protocol for nucleated blood, and the initial DNA concentration was measured by Implen C40 nanophotometer (Implen GmbH, München, Germany).

### 2.2. De Novo Mitogenome Sequencing and Assembly

Mitochondrial genomes (mitogenomes) of the six parrot species in this study were reconstructed by first aligning them to the sequence of *A. vittata* that was sequenced and assembled using three separate approaches: (1) Sanger sequencing using overlapping primers designed on a published sequence of another *Amazona* species [28]; (2) Lifting mitochondrial reads, contigs, and scaffolds from the previously published *A. vittata* genome based on short reads and mate pairs [26]; and (3) identifying long mitochondrial fragments in the whole-genome resequencing of *A. vittata* with the PacBio RS II system [29].

First, we designed primer pairs to amplify and sequence the overlapping regions of the mitogenome of *A. vittata* using the published mitogenome sequence of the closely related yellow-shouldered amazon (*A. barbadensis*) [28]. These primer pairs were designed to obtain overlapping fragments between 500 and 3100 bp in order to sequence and assemble the entire mitogenome and were optimized for amplification and Sanger sequencing across different species. The primer pairs and their PCR conditions are shown in Appendix A.

Second, we searched the previously published de novo genome assembly of *A. vittata* for mitogenome sequences [26]. Three of the scaffolds contained the mitochondrial genome: 2697, 5035, and 2641. Unfortunately, none of them included the control region. We added the corresponding raw reads from the whole-genome assembly and realigned them to the consensus of the Sanger sequences as a separate track using Circos [27] (Figure 1). The average coverage by Illumina reads was 140 reads/bp with a median of 104 read/bp and was highly skewed with the maximum coverage of 728 in the control regions (Figure 1; in orange).

The whole genome of *A. vittata* was also sequenced using the PacBio RS II system [28], yielding around 4 Gb of long reads with 2× coverage. Raw PacBio reads were corrected with LoRDEC 0.5.1 [30] based on the reference set of Illumina reads, using k = 17 and k = 23, with other parameters set as default. The regions that remained weak after correction were outputted in lower case characters and were trimmed, while the solid regions were outputted in upper case characters [30]. PacBio coverage was computed as the total number of nucleotides divided by the expected genome size (1.58 Gb), and this was equal to approximately 2×. We extracted raw mtDNA sequences from the corrected PacBio reads using the Cookiecutter software [31] with 23-mers generated from the draft mitogenome assembly. Cookiecutter extracted 11 reads, and 6 of them were successfully mapped to the draft assembly and covered the whole mitogenome with 1× coverage.

Initially, we assembled a nearly complete mitogenome of *A. vittata* using Sanger sequencing reads from overlapping primers (Appendix A) using the Geneious 8.0.5 software [32]. However, several regions of this assembly remained absent due to the technical difficulties in amplifying and sequencing across the repetitive regions of the control region. Therefore, we applied a combination of different technologies to assemble the complete mitogenome sequence for full annotation. For the short-read assembly, we took Illumina paired-end sequences from the *A. vittata* de novo genome assembly [26] and developed our own naïve approach based on a supervised reference-assisted de Bruijn graph walkthrough, implemented in g language. First, we used Jellyfish 2 [33] to compute 23-mer frequencies in the Illumina raw reads. Afterward, the k-mers were mapped to the closest available high-quality genome (*Melopsittacus undulatus*) (GenBank accession number: NC_009134) with a *kmer_cov_for_fasta.py* script. The first 23-mer with an exact match to the 12S rRNA genes in the *M. undulatus* mitogenome was used as a starting point for extension in both directions. Python scripts *kmer_continue_left.py* and *kmer_continue_right.py* extended a given k-mer with respect to the difference in autosomal and mitogenomic 23-mer coverages until it encountered a bifurcation due to the presence of repeats in the sequence. To resolve the forked path, manual intervention was required: we checked every possible path extension from the bifurcation and selected one that did not disagree with PCR product sizes, Sanger sequences, Illumina paired-end reads, and PacBio reads. This task was feasible because of the small size of the mitogenome, and the relatively low number of repeats encountered, therefore this cannot be replicated on a larger scale. Python scripts are publicly available in the GitHub repository (https://github.com/Applied-Genomics-Laboratory/mtDNA_assembly; accessed on 22 April 2020).

The assembled *A. vittata* mitogenome was verified by three different approaches: (1) assembly consistency between the PacBio long reads and the assembled mitogenome using the Blasr tool [34], (2) the assembly length consistency with PCR product sizes; and (3) consistency with the coverage of the Illumina reads. We have very good confidence in the mitogenome assembly of the Puerto Rican parrot (*Amazona vittata*) (Figure 1). It is important to emphasize that the complete mitochondrial assembly of the *A. vittata* is based on the very long and overlapping PacBio reads (Figure 1, track 2) that span the entire mtDNA genome without gaps, and Sanger and Illumina reads are only aligned to them afterward. PacBio was attempted because PCR did not work well across this region, possibly due to the hairpins being formed by the two nearly identical CR, and therefore Sanger sequencing was not an option. Six long reads with average lengths of 3–4 kb (Figure 1) mapped to the mtDNA genome, and, fortunately for us, one of the PacBio reads from the whole-genome sequence completely included the CR1 and CR2 and their flanking regions and put them in the correct orientation. PacBio has less accuracy in calling the bases than Sanger of Illumina, but it can be corrected using Illumina reads to increase accuracy. Since the raw Illumina reads were generated not specifically for the mtDNA but for the whole-genome assembly, the coverage by the raw Illumina reads is expected to vary, especially because some of these reads could therefore represent NUMTS that would align to the mtDNA.

### 2.3. Reference-Based Mitogenome Assemblies

Once one mitogenome was fully assembled, we used additional sequencing as well as several different published sources to reconstruct mtDNA for other species using *A. vittata* as a reference sequence. First, the mitochondrial genome sequences for *A. ventralis*, and *A. leucocephala* were extracted from the genome data used in previous publications [25,35,36]. Second, the mitogenome of the red-browed amazon (*A. rhodocorytha*) was also sequenced as a part of whole-genome sequencing using Illumina paired-end libraries (Appendix A). Each of the samples was produced following the best practices for NGS data [37]. Finally, for the two Jamaican species, *A. agilis* and *A. collaria*, a 10X Genomics Chromium library was generated using 0.625 ng of DNA as input (https://support.10xgenomics.com/de-novo-assembly/library-prep; accessed on 20 April 2019) in preparation for a full genome assembly (unpublished data). Each library was then sequenced on the Illumina HiSeq X platform, totaling 396M and 379M read-pairs for *A. agilis* and *A. collaria*, respectively. All raw reads in this study were filtered before the mitogenome assembly, so that for all paired-end reads, both reads had at least 90% of bases with a base quality greater than or equal to Q20. The number of reads of all insert sizes for both the original and filtered datasets are listed in Appendix A.

Mitochondrial genome assemblies for *A. ventralis*, *A. leucocephala*, *A. albifrons*, and *A. rhodocorytha* were created by mapping paired Illumina reads to the new assembled mitogenome of *A. vittata* using *bwa mem* [38]. Unmapped reads were filtered out with Samtools [39]. Final assembly from the successfully aligned reads was carried out using Geneious 8.0.5 software [32]. The red-browed amazon (*A. rhodocorytha*) is a South American species that was included to provide an outgroup for our phylogenetic analyses (Appendix A).

To reconstruct the mitochondrial genomes of the two Jamaican amazons (*A. agilis* and *A. collaria*), a BLAST database was created using the NCBI-BLAST-2.3.0+ [40] *makeblastdb* command from the Supernova [41] assembly of paired-end Illumina reads prepared using the 10× Genomics protocol. The *A. vittata* mitogenome sequence was queried against the BLAST database using *blastn* [42] and sequences with the best hits were extracted from the Supernova assembly. Final mitochondrial genome assembly from the successfully aligned reads was carried out using Geneious 8.0.5 software [32].

### 2.4. Mitogenome Annotation

We annotated protein-coding genes (PCGs) in the mitogenome of *A. vittata* by identifying all possible open reading frames (ORFs) with a custom script implemented in Python and queried them against the NCBI nucleotide database. Non-coding tRNA genes were annotated with tRNAscan-SE [43]. For all the remaining species in the study, PCGs and other elements were identified by aligning them to the annotated sequence of *A. vittata* in Geneious 8.0.5 [32].

Nuclear copies of mitochondrial DNA (NUMTs) [44] are a feature that could have additionally contributed to the difficulty with the resequencing of this region. To evaluate this possibility, we searched mitogenomic sequences of *A. vittata* for NUMTs against the previously published genome of *A. vittata* (Kolchanova et al., 2019) [25], and identified 34 candidate sequences, detailed information about which can be found in Appendix B. We identified candidate sequences of nuclear mitochondrial DNA segments or NUMTs [44] as described by Nacer and Raposo do Amaral [45], by performing a BLASTN v2.10.0 [46] search of the complete mitogenome of *A. vittata* against the complete nuclear genome assembly of the same species. Searches were conducted using the command line version of BLASTN. Only hits with an expected value (*e*-value) equal to or smaller than 10^−4^ were considered, and no filters (e.g., low complexity) were used. Two or more identical matches were counted as one. The identity percentage of candidate hits was analyzed before they were considered NUMTs. The threshold was established by calculating pairwise identities between every complete mitogenome from the Greater Antillean *Amazona* species available at the time of this study. The largest identity was 96.8% between *A. vittata* and *A. ventralis*, the two closest species among all six. Anything above this value was considered to be actual mtDNA misidentified as nuclear [45] and such candidates were discarded as false positives (Appendix A).

### 2.5. Mitogenome Sequence Capture for Annotating Diversity within Species

In addition to the mitogenomes generated from the whole-genome Illumina paired-end methods described above, we resequenced 20 additional samples for population analysis that are listed in Appendix A, namely: three *A. agilis* and seven *A. collaria* individuals were chosen from a population of rescued birds of different ages at Hope Gardens in Kingston Jamaica (2015), as well as 10 unrelated *A. vittata* fledglings, hatched at the Conservation Program of the Puerto Rican Parrot, USFWS, Puerto Rico in 2016.

To resequence only the mtDNA of these individuals, we first captured the desired fragments using molecular baits. Specifically, we used the publicly available *A. ventralis* reference mitogenome sequence (GenBank^®^ Accession: NC_034679.1), to custom design probes to capture and resequence multiple individuals. The probes were designed using MyBaits^®^ at Arbor Biosciences™ based on a bait length of 80 bp at 4x tiling according to the manual [47]. The Baits are now available as a predesigned MyBaits^®^ Mito panel on the company website (https://arborbiosci.com/genomics/targeted-sequencing; accessed on 15 February 2019). The captured fragments from the 20 individual samples were subsequently sequenced on the Illumina HiSeq X Ten platform, and reads were aligned to the reference mitochondrial genome of *A. vittata* with *bwa mem* [38].

The final assembly from the successfully aligned reads was performed using Geneious 8.0.5 software [32]. The consensus sequences were aligned using Kalign (Penalties: Gap open-80, gap extension 3, terminal gap-3) [48]. Disagreements with the *A. agilis* sequence were exported into a .csv file using Unipro UGENE software [49]; 100-block counts of the disagreements were calculated with a custom script written in Python programming language using Pandas data science library (https://pandas.pydata.org/; accessed on 22 April 2020). Intra and interspecific genetic differences were illustrated using Circos plots [29].

### 2.6. Multiple Alignments and Phylogenetic Analyses

We used maximum likelihood (ML, Appendix A), maximum parsimony (MP, Appendix A), and Bayesian inference (Appendix A) to reconstruct phylogenetic relationships and estimate divergence times among *Amazona* parrots from the Greater Antillean islands. Along with the mitogenomes newly generated for this study, we used previously published mitogenomes from other *Amazona* species as well as 10 other neotropical parrot species (subfamily *Arinae*) (see Appendix A). There was a total of 33 individual parrot mitogenomes, belonging to 17 different species.

For the phylogenetic analyses, we included sequences from the 13 protein-coding genes (PCGs), 22 tRNAs, and two rRNAs. The control region sequence was excluded from the analyses as it was too variable for the interspecific scale, and since no long reads existed allowing independent assembly for each species, it could potentially contain indels resulting from the reference mapping bias. Multiple sequence alignments containing all the partitions from the 33 individual parrot mitogenomes used in the analysis were generated using MAFFT v7.017 [50] in Geneious 8.0.5 [32] with the following parameter settings: scoring matrix = 200, PAM/k = 2, gap open penalty = 1.53, offset value = 0.123. Alignments were then manually inspected for possible inconsistencies.

Maximum parsimony (MP) analysis was conducted using MEGA X software [51,52]. This analysis only included six mitogenome sequences representing each of the five amazon parrot species from the Greater Antilles and *Amazona albifrons* designated as the outgroup. The MP tree was created using the Subtree-Pruning-Re-grafting (SPR) algorithm with the starting trees obtained by random addition of sequences (10 replicates). Node support was evaluated by bootstrapping using 500 replicates. Absolute numbers of nucleotide changes in the branch relative to the common ancestor were noted along phylogenetic lineages (Appendix A). The MP tree had a length of 1924 steps with a consistency index of 0.685), a retention index of 0.621, and a composite index of 0.538 for all sites (or 0.425 for 3579 parsimony-informative sites).

To estimate the maximum likelihood (ML) phylogeny, we partitioned the mitogenome sequences into PCGs, tRNAs, and rRNAs using PartitionFinder2 [53,54], and used W-IQ-TREE [55], a web interface and a server, IQ-TREE, version 1.6.9 [56]. The nucleotide substitution models for the three data partitions were selected automatically using ModelFinder [57,58] and the Bayesian information criterion prior to the tree search (setting: Substitution model: “Auto”). The tree search parameters included a perturbation strength of 0.5 and an IQ-TREE stopping rule of 100 (default settings). Branch support was estimated with a standard bootstrap analysis using 1000 bootstrap iterations. Output trees were visualized in FigTree v1.4.3 [59] (Appendix A).

For the Bayesian inference, we used BEAST version 2.3.1 [60] to co-estimate the phylogeny and divergence times, employing the three data partitions of the mitogenome dataset (PCGS, tRNAs, and rRNAs). An .xml file was created using the BEAUTi application. The best-fitting nucleotide substitution models were selected using the Bayesian model test package, bModelTest implemented in BEAST 2.3.1, which uses a reversible jump algorithm to switch between substitution models over the nucleotide data, with or without γ rate heterogeneity and/or with or without invariant sites [61]. Substitution site models were unlinked, base frequencies were set to “Empirical”, a strict clock model was assumed, and the Yule process of speciation was set as the tree prior. Due to the scarcity of the fossil record for the predominantly tropical Psittacidae clade, it was not possible to include primary calibration priors for our phylogenetic analysis, particularly for the Greater Antillean island Amazona parrots. Consequently, we employed three secondary calibration priors (node ages inferred by other molecular phylogenetic studies based on primary fossil data). The secondary calibration priors were adopted from Rheindt et al., [62]. Specifically, these included the *12.12* MYA split between *Ara* and *Aratinga*, and the 23.48 MYA split between *Amazona* and (*Ara* + *Aratinga*). Finally, the divergence time between the genera *Amazona* and *Pionus* was estimated to have a median of ~13 MYA (4.5–22.4 MYA) based on data reported earlier [63,64], and this was used as a third calibration prior. Two separate MCMC analyses were run for 20,000,000 iterations, with trees and parameters sampled every 2000 generations and the first 10% of these discarded as burn-in. Posterior distributions and ESS values of tree likelihoods, substitution, and molecular clock parameters from each run were checked using Tracer [65]. The post-burn-in samples of the posterior distribution were merged using LogCombiner [60] and TreeAnnotator [59] was used to prepare the maximum clade credibility topology with mean node heights for visualization in FigTree v1.4.3 [59] (Appendix A). The same analysis was repeated using an uncorrelated relaxed clock model (Appendix A), where evolutionary rates at each branch were drawn from a log-normal probability distribution [66].

### 2.7. Reconstructing Dispersal and Speciation

Biogeography has a number of methods for inferring ancestral geographic ranges on phylogenies, many of which use different, often conflicting, assumptions. We used the R package *BioGeoBEARS* to infer the biogeographical history as multiple biogeographical models can be tested [67,68] given the observed data from a phylogenetic analysis combined with geographical data, such as sizes and distances between landmasses.

Specifically, the *BioGeoBEARS* package implements several models based on different statistical approaches, including DIVALIKE [69,70], DEC [71,72], and BAYAREALIKE [73]. DIVALIKE is a Dispersal–Vicariance analysis (DIVA) implemented in the program DIVA [74]. This is a parsimony method in which the ancestral area with the least cost is chosen as the most optimal. While this program assigns a cost of one for both colonization and extinction, it assigns no cost for vicariance and within-area speciation events [69]. Therefore, this approach maximizes vicariance explanations and minimizes the number of dispersal events. DEC or The Standard Dispersal–Extinction–Cladogenesis is a likelihood-based approach that calculates the biogeographic likelihood of a phylogenetic tree using the range data arrayed at its tips [75]. The DEC model assigns probabilities of range transitions as a function of time using two free parameters: one specifying the rate of dispersal or range expansion, and the other specifying extinction or range contraction [72]. Finally, the BAYAREALIKE represents a Bayesian approach for inferring speciation and biogeographic history that involves a large number of areas [73]. Here, the tree is first populated with a history of biogeographic events that are consistent with the observed species ranges at the tips of the tree. Then, the likelihood is calculated for each given history accounting for the exponential waiting times between biogeographic events and the relative probabilities of each biogeographic change [73]. The *BioGeoBEARS* package also implements the “+J” versions of these models [68,76]. which include founder-event speciation. All three of these (DIVA + J, DEC  +  J, BAYAREA + J) add a possibility of founder-event speciation denoted by a new free parameter, J⁠ [68]. Since there is no simple common statistical framework by which to judge which models are preferable, we present the results of all possible models and include the interpretation of each (Appendix A).

All models were tested with two time periods (Appendix A). The first period is characterized by the connection between the Central American Mainland and Jamaica (MJ) via the emergent Nicaraguan Rise—a wide triangular ridge that extends from Honduras and Nicaragua to Hispaniola that has submerged in Pliocene [77]. During Pleistocene, the sea level was low, so the island of Jamaica was much closer to Central America via the exposed Nicaraguan Rise—a wide triangular ridge that extends from Honduras and Nicaragua to Hispaniola, that allowed active dispersal from the mainland to Jamaica. The stepping-stone had probably been present until the eventual drowning sometimes in the Pliocene [78]. This is why we added the second time period (after ~3.3 MYA) where the models assume current distances between the islands and no direct connection between Nicaragua and Jamaica. After this time, the closest landmass to the Central American mainland is now Cuba, not Jamaica (Appendix A).

## 3. Results

### 3.1. Mitogenome Sequencing, Assembly, and Annotation

The graphical representation of the *A. vittata* mitochondrial DNA assembly and annotation is shown in Figure 1. Mitogenome sequencing was conducted in three consecutive stages, using different technologies, each complementing and extending the previous results. The first attempt was to design overlapping primer pairs (Appendix A) and sequence the mitogenome using Sanger sequencing based on the available mitogenome of *A. barbadensis* [27]. Unfortunately, the primer pairs do not work across the duplicated control regions (CR1 and CR2, Figure 1). This fragment was amplified and verified according to the lengths of PCR products, but Sanger sequencing across the region was not able to resolve the sequences of CR1 and CR2, possibly because of heteroduplexes forming between the two highly similar regions. A similar issue was also previously reported in *A. barbadensis* by Urantowka et al. (2013).

To resolve this issue, we first tried to reassemble the mitogenome using data from Illumina paired-end sequences from the *A. vittata* de novo genome assembly (Appendix A). During the assembly of these data, it became evident that the duplicated CR1 and CR2 regions were GC-rich and low-complexity, as well as too wide apart to be bridged by the Illumina paired-end reads or contigs assembled earlier. Repeats in these regions are known from other species of Amazons [27] and are not unique to birds [79].

Due to these complications, we attempted to search the low-coverage PacBio whole-genome data for long reads that would span across the entire region of the duplicated control region. The *Cookiecutter* software [31] extracted 11 reads, 6 of which were successfully mapped to the draft *A. vittata* assembly and provided 1x coverage of the mitogenome (Appendix C). One of the PacBio reads in this set extended across the entire length of CR duplication and enabled us to obtain a complete assembly of the *A. vittata* mitogenome (Figure 1). This long read assembly was further supplemented by Illumina reads from other studies and by the overlapping Sanger reads from this study that were used to correct the long PacBio reads.

The complex nature of the duplicated control region and tandem repeat regions prevented us from using the traditional Sanger sequencing approach to amplify and sequence this region using overlapping fragments in *A.*
*vittata*. To overcome this complexity, we used a combination of several next-generation sequencing technologies, producing a hybrid mitogenome assembly for this species. In this study, we have found that mitochondrial genomes of all representatives of the Greater Antillean amazons contain the nearly exact duplicated control region (CR) and a gene arrangement that is characteristic for all other amazon parrot mitogenomes analyzed so far [80,81].

The annotation of the mitochondrial genome of *A. vittata* is shown in (Figure 1) with protein-coding genes, rRNA genes, tRNA genes, control region, tandem repeats, and GC content shown as different tracks. In total, we annotated 23 tRNAs and tRNA pseudogenes, 14 protein-coding genes and pseudogenes, the duplicated control region, repetitive sequences, and two rRNAs. In addition, a separate annotation figure displaying locations of the nuclear copies of mitochondrial DNA (NUMTs) is presented as Appendix A [44]. These segments range from 54 to 5499 bp, with only five longer than 1000 bp with a mean of 604.4 bp ± 993.9 (standard deviation). The total length of NUMT sequences is 20,550 bp in 1,446,972,151 bp or 0.00142% of the nuclear genome, and their identity to the original mitochondrial sequence ranges between 64.8% and 95.2%. While these nuclear copies should not have complicated the assemblies, two of the NUMTs overlap with the CR1 and CR2 regions (Appendix A) and may explain the large number of Illumina reads aligned to these regions (Figure 1, track 1), which has at first interfered with the short read assembly, therefore requiring the long read (PacBio) inclusion (Figure 1, track 2).

The 34 of NUMTs we found in *A. vittata* is well within the range reported in other neotropical parrots, which is around 30, but the number is larger than 27 as discovered previously by Liang et al. [82] in the earlier version of *A. vittata* genome [26], which is likely due to the older draft assembly being less complete that the one used in this study [25,83].

### 3.2. Phylogeny and Evolutionary History

The parrots primarily found in South America form a clade distinct from those in Central America and the Greater Antilles archipelago. Based on our Bayesian inference analysis, this split was estimated to have occurred approximately 5.94 MYA (95% highest posterior density [HPD] = 4.42–7.71 MYA) (Figure 2, for full phylogeny with all individuals and with strict and relaxed clock models see Appendix A). The five amazon parrots from the Greater Antilles form a separate clade that split from the mainland Central American white-fronted amazon (*A. albifrons*), approximately 3.43 MYA (HPD = 2.52–4.46 MYA).

*A. vittata* endemic to Puerto Rico and *A. ventralis* from Hispaniola were resolved as sister species, and last shared a common ancestor only 0.69 MYA (HPD = 0.49–0.9 MYA). These two species shared common ancestry with *A. leucocephala* from Cuba as recently as 0.76 MYA (HPD = 0.55–1.01 MYA). The majority of nodes (indicated in Appendix A by numbers 1–16), had 100% bootstrap (maximum likelihood), or 1.0 posterior probability (Bayesian inference) support. The only notable exception was the clade containing *A. ventralis* + *A. vittata* (Figure 4, node 8), which received posterior probability support of 0.84 and bootstrap support of 88%. The weak phylogenetic signal associated with this clade is consistent with the low sequence divergence (3.2%; 96.8% sequence identity) and recent inferred age of the split (0.69 MYA) between these two species. The separation between these two species and *A. leucocephala* in Cuba involves an extremely short internal branch, suggesting a near-polytomy in their relationship.

To confirm these findings, we also constructed an ML tree (Appendix A) and an MP tree (Appendix A), depicting absolute numbers of nucleotide changes relative to the common ancestor along phylogenetic lineages. The tree topologies are congruent with the one obtained using Bayesian inference (Figure 2 and Appendix A). The ML and MP trees also support the most likely phylogenetic relationships between the species of interest: *A. vittata* and *A. ventralis* are sister species, and the Cuban *A. leucocephala* is sister to both. Among the Jamaican parrots, an earlier split is observed for *A. agilis*, which is basal to all the other Greater Antillean amazons, and *A. collaria* is more closely related to *A. leucocephala*, than to *A. agilis*.

Overall, our phylogenetic analyses suggest successive speciation of amazon parrots during the stepwise colonization of each island in the Greater Antilles. The two species of the Amazons endemic to Jamaica, *A. agilis* and *A. collaria*, were resolved as successive sister lineages, suggesting that these species arose independently. Another potential interpretation could be the initial colonization of Jamaica, followed by a split into two subpopulations, one of which gave rise to *A. agilis*, and another—to *A. collaria*. The latter may have also first dispersed to Cuba to establish the founder population of *A. leucocephala*. To evaluate different speciation scenarios, we followed phylogenetic analysis with the biogeographic analysis of dispersal and speciation using the *BioGeoBEARS* package [67].

### 3.3. Dispersal and Speciation

A number of different models were tested to infer possible speciation and dispersal routes using the *BioGeoBEARS* package [67] including three approaches: parsimony (DIVALIKE) [69,70], Maximum Likelihood (DEC) [71,72], and Bayesian (BAYAREALIKE) [73] (Appendix A) using a distance matrix based on the sizes and the shortest distances between the landmasses and islands (Appendix A). In addition, the models were modified with an additional parameter (J) to evaluate the possibility of founder-event speciation (DIVA + J, DEC  +  J, BAYAREA + J) [67]. Each model resulted in a slightly different speciation and dispersal history therefore we included model descriptions and interpretations of the simulations for each (Appendix A).

The simulation based on the DIVALIKE model (DIVALIKE + X; Appendix A) suggests that there was a shared gene pool extending across Central America and Jamaica (MJ) before the drowning of the Nicaraguan Rise. When the rising sea water severed the connection, the shared gene pool was split in two. *Amazona albifrons* then continued to evolve on the mainland and Yucatan peninsula (MY), while the population in Jamaica was split again. The resident population eventually evolved into *A. agilis*, while some individuals dispersed to Cuba (C) to establish a founding population of *A. leucocephala*. Sometime after this split, the Cuban population dispersed to Hispaniola (H) and quickly split into two separate lineages: *A. ventralis* on Hispaniola and *A. vittata* in Puerto Rico. However, for approximately 1 MYA there is another dispersal of individuals from Cuba that return back to Jamaica to establish the foundation of the future *A. collaria* population.

A version of this model that allows a possibility of founder-event speciation (DIVALIKE + J; Appendix A) produces slightly different results. The first difference is that the original founder population in Jamaica quickly splits into two. This model suggests that for some time, there are two species cohabiting this island. One of them will establish the lineage leading to the extant *A. agilis*, while the other is to become the common ancestor of *A. collaria*. This scenario would imply that some of the migrants from the latter lineage subsequently dispersed to Cuba (*A. leucocephala*). After this, the dispersals between Cuba, Hispaniola, and Puerto Rico proceed in a stepwise manner, without forming any intermediate shared gene pools (Appendix A). Other variations (DIVALIKE2+X and DIVALIKE 2+J), allowing extinction estimates, produce results that were identical to DIVALIKE + J (Appendix A). On the other hand, the simulation based on the DEC + X model (Appendix A) is identical to the DIVALIKE model (DIVALIKE + X; Appendix A) and its modified version, allowing for the possibility of founder-event speciation, is also very similar to the DIVALIKE model where the of *A. collaria* is established by dispersal from Cuba (DEC + J; Appendix A). The addition of the “J” parameter results in small modifications of the resulting model: (a) it incorporates Yucatan along with the South American Mainland (MY) into the common ancestral area, and (b) it does not require a temporary shared genetic pool for the ancestral Cuba and Jamaica that was present in DEC + X model (Appendix A).

A simulation based on the BAYAREALIKE + X model (Appendix A) suggests the existence of an original shared genetic pool in Central American Mainland and Jamaica (MJ) before drowning of the Nicaraguan Rise, with a subsequent split into two—one in Yucatan that continued to evolve into the modern *A. albifrons* and another in Jamaica that would give rise to all the other lineages. However, unlike all of the other models, this model suggests that the next speciation followed the dispersal from Jamaica to Hispaniola. A common gene pool that existed between these two islands (JH), which eventually fractured. The population remaining in Jamaica eventually evolved into *A. collaria*. From the remaining population in Hispaniola that eventually evolved into *A. ventralis*, founders dispersed and established themselves in Cuba and Puerto Rico (*A. leucocephala* and *A. vittata*). The addition of the J parameter to this model (BAYAREALIKE + J; Appendix A) makes the outcome very similar to the results of DIVALIKE2+X and DIVALIKE2 + J (Appendix A). The main features of the eighth models used in our studies are combined in Appendix A.

### 3.4. Mitogenome Diversity

The mitochondrial genomes we assembled for the Greater Antillean parrot species in this study (*A. ventralis*, *A. leucocephala*, *A. agilis*, *A. collaria*, and *A. albifrons*) were all aligned to the *A. agilis* mitogenome (Figure 3). *A. agilis* was chosen as a reference for this visualization since this way it is easier to see the diversity within the island clade (Figure 3, tracks 2–5) compared to the differences versus the mainland species represented by *A. albifrons* (Figure 3, track 1). Figure 3 displays only the differences between species, the intraspecific variation is shown in Figure 4.

The interspecific diversity among the *Amazona* spp. of the Greater Antilles is not evenly distributed across the mitochondrial genome. The most striking feature is the markedly distinct accumulation of changes associated with the flanks of the duplicated control region, especially tandem repeat regions 1 and 2 (TR1 and TR2 in Figure 3). This could potentially be due to errors during the assembly of highly repetitive sequences in these loci. On the other hand, it can be clearly seen that fractions of both CR1 and CR2 are highly conserved relative to the other regions in the mitogenome, as we observe the least amount of between-species variation. These regions are thought to be involved in mtDNA replication and transcription [80]. Therefore, significantly large regions of the two copies within the same genome also remain nearly identical, which adds complexity to the assembly process. Figure 3 additionally shows several other regions of low interspecific diversity, as well as regions of high divergence (such as *CYTB*, *ND2*, and *ND6*), found outside of the D-loop.

To illustrate the distribution of differences within populations of certain species of the Greater Antillean parrots, multiple individuals from four of the five species were sequenced. Specifically, we sequenced the mitogenomes of 10 unrelated *A. vittata* fledglings hatched at the Conservation Program of the Puerto Rican Parrot, USFWS, Puerto Rico in 2016, as well as two *A. ventralis* individuals from the same program, two *A. agilis,* and seven *A. collaria* individuals, chosen from a population of rescued birds of different ages at Hope Gardens in Kingston Jamaica (2015). Averages of the number of intraspecific variants per 100 bp were calculated with a custom python script and displayed in a Circos plot (Figure 3).

This analysis indicates a distinct accumulation of changes associated with the flanks of the control region duplication, similar to what was shown between the species (Figure 3). At the same time, this analysis demonstrates very low diversity in *A. vittata* compared to the other species (Figure 4, innermost circle): only three variable regions are evident in this species with only 1–2 differences specific to individual birds. All birds were identical in their mitochondrial sequence outside of the TR fractions of the duplicated control region. It should be noted that the observed differences in the TRs could have arisen from assembly errors due to the low complexity of these regions. In our analysis, *A. vittata* had the lowest mitogenome diversity even though the largest number of individuals were sequenced. Our data also indicate low mitogenomic diversity in *A. agilis* from Jamaica (Figure 4, outermost circle). Only three variable regions were identified between the two birds, which may be due to the small sample size used for the comparison. However, mitogenomes of the two *A. ventralis* individuals demonstrate much higher diversity that is not concentrated in the repetitive regions. The level of intraspecific diversity observed in the seven mitogenomes of *A. collaria* in this analysis is also higher than that of *A. vittata* and *A. agilis*.

The resulting assemblies were inspected for patterns of genetic diversity both among species and within species where possible. We had expected that the TR1 and TR2 regions of the CR1 and CR2 would be the most variable, which is also evident from Figure 3. At the same time, the duplicated CR feature also includes a region in the middle of each repeat known to be conserved and, as already mentioned earlier, is thought to be involved in the crucial processes of mtDNA replication and transcription [80].

## 4. Discussion

Amazon parrots in the Greater Antilles represent a fascinating model of speciation on islands, similar in many ways to that of Darwin’s finches in the Galapagos [84,85]. *Amazona* spp. parrots are a diverse group distributed throughout the Neotropical region. Of the 30 known extant species, eight are currently found in the Caribbean, representing at least three different colonization events from mainland Central or South America [16]. Currently, each species (except for *A. leucocephala*) is restricted to a single island and has been subject to adaptive and non-adaptive forces to evolve distinct phenotypes (feather coloration patterns, calls, nesting behavior, etc.).

In this study, we addressed the diversity, phylogeny, biogeography, and conservation of the *Amazona* species endemic to the islands of the Greater Antilles as well as one South American species using fully assembled mitogenomes, rather than the fragments amplified from museum samples. We combined the information from the newly sequenced mitogenomes (Appendix A), with the publicly available full mtDNA sequences from other neotropical parrots (Appendix A), to make inquiries about the evolutionary history and genetic diversity of the Caribbean parrots that was not possible with the smaller fragments. Our taxon sampling of all extant species of *Amazona* parrots from the Greater Antilles allowed us to evaluate specific hypotheses about island colonization and speciation that had been proposed earlier [15,17] (Appendix A). As a result, we contribute new information on the structure of the mitochondrial genome (Figure 1), interspecific (Figure 3) and intro-specific diversity (Figure 4), phylogeny and timing of speciation (Figure 2), and evaluate possibilities of various dispersal and speciation scenarios of the Amazon parrots in the Greater Antilles (Figure 5).

### 4.1. Mitogenome Diversity

Chromosome-level assemblies hold the ultimate potential to reveal answers to all kinds of evolutionary questions (e.g., evolutionary history of genes, gene families and of the species themselves, past population dynamics, more precise times of speciation events and phylogenetic relationships with other species, understanding gene functions and the evolution thereof, etc.) [86]. On the other hand, the computational complexity of most of these problems is still immense, and even the task of accurate de novo whole-genome assembly is not yet completely solved [87,88]. Nevertheless, in many cases, assembly of every single chromosome is not necessary to address many important questions about the species’ evolutionary history, and mitogenomic data can help expand the body of knowledge on parrot genomics and conservation genetics, albeit this represents only a single gene tree due to the non-recombining nature and maternal inheritance of this organellar genome [89].

The complete mitochondrial sequences from this and other studies [90] allowed us to construct detailed annotated maps of the mitochondrial DNA and its diversity (Figure 1), take a first look at the divergence between all extant Amazon parrots in the Greater Antilles (Figure 2 and Figure 3), and provide mitogenomic diversity using full mitogenomes in four of these species: *A. vittata, A. ventralis, A. agilis,* and *A. collaria* (Figure 3).

Among all species in this study, *A. vittata* has the lowest number of differences (34) in the mitochondrial DNA despite having the most individuals (10) sequenced (Figure 4, Appendix A). A slightly higher number of variants was reported for *A. agilis* (112). All these differences are localized in the control region and none of them are in the coding regions (Figure 4, Appendix A). This exceptionally low genetic diversity conforms to the endangered status of A. vittata and points to a possible conservation concern for *A. agilis*. All mitogenome differences are reported in Appendix D and visualized in Figure 2, Figure 3 and Figure 4. We developed working primer sets that can be used to amplify and focus on specific regions harboring genetic diversity in one or more species (Appendix A).

### 4.2. Phylogeny

Our phylogenetic analyses based on near-complete mitogenomes (excluding only the duplicated control region) resolved phylogenetic relationships among *A. vittata*, *A. ventralis*, and *A. leucocephala* with moderate nodal support. This node was not fully resolved in the previous study that used 3160 bp distributed across three mitochondrial and three nuclear gene fragments [16]. The previous alignment of the six data partitions for 48 taxa contained 560 parsimony-informative sites, while in our study with 33 taxa (including multiple individuals for several species), the mitogenomes contained 3579 parsimony-informative sites. Consistent with other studies (e.g., [80]), the high information content of mitogenomes, although representing only a single gene tree, provides sufficient resolving power even when taxon sampling density is high.

The Bayesian analysis allowed for the reconstruction of topologies that were consistent with the prior knowledge about phylogenetic relationships and node ages [25], as well as with geological data [91,92,93,94]. The topology of the Bayesian tree was also fully consistent with the trees acquired using maximum likelihood and maximum parsimony, Appendix A).

The fossil record of neotropical parrots is scarce. Birds generally do not fossilize very well: their bones are low in density, fragile, and more likely to break down rather than fossilize. In addition, animals from humid tropical forests are rarely preserved, because they decay rapidly in those environments. This did not allow us to use primary (fossil) calibrations and we had to use secondary priors for our divergence dating analyses.

Nevertheless, our posterior (estimated) divergence times for Caribbean *Amazona* are in agreement with the previously proposed hypotheses concerning their diversification and its timing [16,20]. Moreover, the dates are consistent with the geological record of the formation of these islands (discussed above): all the inferred parrot speciation events postdate estimated emergence times for the islands and other key geological events in the region, the appearance of potential migration routes, and ecological opportunities following the closure of the Isthmus of Panama, and submergence of the Nicaraguan rise [91,92,93,95,96,97].

It has been postulated that the origin of the entire West Indian avifauna was by dispersal, largely from North America [98]. Previously proposed dispersal-speciation scenarios based on morphological data suggested a close relationship between *A. collaria* and *A. leucocephala* [15,17] (Appendix A), which has been further corroborated by molecular data [16,20]. The scenarios disagreed on the sequence of the dispersal and speciation events. We used phylogenetic analysis to try to reconstruct the sequence timing of dispersal and speciation events that would explain the current distribution of parrot species on the Great Antillean islands (Cuba, Jamaica, Hispaniola, and Puerto Rico).

According to our estimates, the split between the South American and Central American amazons occurred sometime around 6 MYA (Figure 2 and Appendix A), as parrots migrated northwards across the landscape of an emerging landmass that led to the complete closure of the Central American Seaway and formation of the Isthmus of Panama in the late Neogene [93,95,96,99,100]. Why this did not happen earlier is not clear but could be caused by parrots’ inability to fly over long distances without feeding, inhospitable landscapes of the emerging land, as well as restricted ecological opportunities when encountering resident Central American fauna [96,101]. While many terrestrial species in the Caribbean are of the vicariant origin [91,102,103] or dispersed along the paths of the prevailing water currents [104] and hurricanes [105], birds can disperse between islands where the distance permits a direct flight. This was possibly facilitated by the low sea levels at the time, opening up opportunities: the birds spread across Central America relatively quickly, and the first round of island colonization in the Greater Antilles probably occurred after 3.47 MYA (Figure 2, Appendix A).

Our analysis also allowed us to resolve the order of colonization in the branch containing species from Cuba, Jamaica, and Hispaniola. According to Lack [17], both *A. ventralis* and *A. vittata* descended from a most recent common ancestor with *A. leucocephala* as a result of direct colonization from Cuba (Appendix A). Alternatively, *A. vittata* was hypothesized to have descended from an ancestor in Jamaica and to have shared the most recent common ancestor with *A. agilis* [15] (Appendix A). Other possibilities both assumed Jamaica to be the stepping stone for the colonization of the other three islands (Appendix A). According to our model, *A. leucocephala* split from the common ancestor it shared with *A. vittata* and *A. ventralis* slightly earlier (0.76 MYA) compared to when the two latter species diverged from one another (0.69 MYA) (Figure 2, Appendix A).

### 4.3. Biogeography

Most of the *BioGeoBEARS* [67] models (6 out of 8) tested to infer possible speciation and dispersal routes agreed that there was an original MJ gene pool connecting the Central American mainland (M) to Jamaica (J) (Figure 5, Appendix A). During the Pleistocene, the sea level was low the island of Jamaica was much closer to Central America via the exposed Nicaraguan Rise, facilitating active dispersal. Most likely, the first speciation event occurred when the Jamaican lineage split from the common ancestor with *Amazona albifrons* that lived on the South American Mainland (Appendix A) and, possibly in Yucatan (MY) (Appendix A) (Appendix A). The stepping-stone path of dispersal has probably been present until the eventual drowning, sometimes as late as the Pliocene [78]. Subsequently, the ancestral population *A. albifrons* remained there, while the Jamaican population was cut off by the rising sea and subsequent drowning of the remnants of the Nicaraguan Rise eliminating any further gene flow. Most of the alternative scenarios from the literature are also compatible with this scenario (Appendix A).

Jamaica is the only island in the Greater Antilles with two extant species of amazons: *A. agilis and A. collaria*. The differences between them are substantial, and there is no evidence that they can interbreed. Our analysis shows that *A. agilis*, represents the earliest independent lineage that split from the common ancestor with *A. albifrons* in the Central American mainland and spread directly to Jamaica around 3.47 MYA (HPD = 3.29–4.03 MYA) (Figure 2 and Appendix A). This time only slightly postdates the low sea level minimum of 4.2 MYA [106,107] which would expose the Nicaraguan rise to the greatest extent [99,100]. It was suggested that the original lineages migrated directly to Jamaica (Appendix A) [15,16,20], using the then-emergent remains of the drowning Nicaraguan rise as a stepping-stone path [91,92,97] and only then did founders migrate to Cuba and other islands (Appendix A).

The “island” lineage did not undergo another split until approximately 3.11 MYA (HPD = 2.29–4.03 MYA) when ancestors of *A. collaria* split from all the other lineages (Figure 4 and Appendix A). The models give different scenarios for the next speciation event. Four models suggest that the linages leading to the two extant species in Jamaica formed on this island and then one of them, *A. agilis*, dispersed to Cuba 1.38 MYA (HPD = 1.01–1.81 MYA) (Appendix A). These models present a sympatric scenario (Appendix A) that is theoretically possible but is specifically not supported by morphological and genetic data in Jamaican amazons [14,15,16,17,20], neither is there much evidence for sympatric speciation in island birds overall [108,109]. One of the possibilities is that speciation was allopatric but has occurred on some of the landmasses that do not exist today, possibly as part of the Nicaraguan Rise, and could not be included in the mathematical models we employed in the current analysis.

Alternatively, two models suggest that the lineage leading to *A. agilis* and other species briefly form a common gene pool between Cuba and Jamaica (CJ, Appendix A). Finally, one model suggests that the speciation originally occurred allopathically after dispersal from Jamaica to Cuba, and only then was the *A. agilis* population established when some individuals returned to establish a new population in Jamaica (Appendix A).

The current distribution of the *A. leucocephala* seems to suggest multiple and continuous dispersals from Cuba, including the possibility that at some point *A. collaria* has returned from Cuba to Jamaica. In addition to the population on the island of Cuba itself (*A. l. leucocephala* (Linnaeus, 1758)), there are several extant subspecies all occupying smaller islands and archipelagos: *A. l. bahamensis* (H. Bryant, 1867), also called the Bahaman amazon that survives in two populations in the Bahamas, *A. l. caymanensis* (Cory, 1886), also called the Grand Cayman amazon, and *A. l. hesterna* (Bangs, 1916) restricted to the island of Cayman Brac [110]. The Cayman Islands are approximately as far from Cuba as Jamaica, and the presence of two subspecies demonstrates the potential for dispersal at this distance. Sequencing and analysis of the genomes of the Jamaican amazons along with *A. leucocephala* subspecies in the Cayman Islands archipelago (*A. l. caymanensis* and *A. l. hesterna*) could also provide further clues regarding the timing and the dispersal paths.

The most likely reason for the smaller islands to have the subspecies of *A. leucocephala* today can be explained by the classic theory of island biogeography [2,6]: the higher rates of extinction on the smaller islands are followed by repeated colonization from the bigger island. In other words, of all the dispersals that originated in Cuba, the possible dispersal of the ancestors of the *A. collaria* back to Jamaica (Table 1, Features 5,6, Appendix A) may have been the one that has survived until this day.

There is also a possibility that current data do not allow us to evaluate the possibility that the initial dispersal of the ancestors of *A. leucocephala*, *A. collaria*, *A. ventralis,* and *A. vittata* was via Cuba (Appendix A) whose outline was roughly shaped during the late Miocene or early Pliocene [91,111]. Earlier dispersal to Cuba may have been less likely because of the greater width of the Yucatán Channel separating Central American and Western Cuba [99,100,106,107], but the dispersal between 3 and 2 MYA seems possible, given the shorter distance from Cuba to the Central American mainland (150 vs. 350 km), with the Nicaraguan rise previously connecting Jamaica likely already submerged [20,77].

Most models agree that once the parrots reached Cuba, they have continued to disperse to Hispaniola and then to Puerto Rico in a stepping stone fashion (Appendix A). Alternatively, one model suggests that Cuba and Hispaniola may have had a temporary shared genetic pool, that quickly split (Appendix A). The extremely short internal branches with the *A. leucocephala + A. ventralis + A. vittata* clade suggest that once these parrots reached Cuba, the lineage quickly spread across Cuba, Hispaniola, and Puerto Rico and later differentiated within each island into a separate species due to geographical isolation. These events are in accordance with the earlier models [16,17,20]. However, the BAYAREALIKE model predicts a shared gene pool between Jamaica and Hispaniola and suggests Hispaniola as the source of colonization of Cuba and Puerto Rico (Appendix A). This model is similar to the scenario initially proposed by Snyder et al. [15].

There may have been other dispersal events between the islands that followed, but only two of these (*A. ventralis* by spreading to Hispaniola around 0.76 MYA, and of *A. vittata* to Puerto Rico around 0.69 MYA) can be traced with the mitochondrial lineages (Figure 2). This also does not preclude the possibility that earlier populations had been established on Hispaniola and Puerto Rico. Genomics signatures of these earlier migrants may be further uncovered when the whole-genome data of these species are analyzed, but early data show a correlation between effective population sizes of species from Cuba and Hispaniola, which can be explained by continuous gene flow between the islands [25,83].

### 4.4. Conservation

Knowledge of evolutionary history is the cornerstone of conservation studies, without it, there is no context to the conservation effort. Ecological and evolutionary processes may act at overlapping time scales, therefore the neglect of evolutionary history in conservation is unwarranted, but unfortunately, evolutionary processes are often overlooked by biologists and decision-makers interested in protecting endangered species [112]. Genomic data and knowledge about the evolution of species of interest provide valuable information that can be essential for conservation management programs [24,113]: past evolution can explain the vulnerability of species to threats (or their ability to adapt and survive) [114] and help identify geographical areas that are likely to be essential for future persistence of biodiversity [115]. Reconstruction of evolutionary histories allows for guiding conservation efforts by defining evolutionarily significant units (ESUs) [116,117] and has already helped augmentation projects to restore genetic diversity, with perhaps the most famous case being the restoration of genetic diversity in the Florida panther based on the knowledge of historic gene flow [118,119].

The varied outcomes of evolution in island settings can demonstrate a great deal about how natural processes have built biological diversity through the formation and divergence of species [5]. The size of an island is probably one of the most important determinants of genomic diversity [120], but human activity has recently become a far more powerful contributing factor affecting wildlife populations. Most of the *Amazona* species that are the focus of this study are listed as endangered, threatened, or vulnerable according to the IUCN Red List of Threatened Species [12,121]. The comparison between full mitochondrial genomes of 10 *A. vittata* parrots which were chosen to be the least related based on the pedigree record kept by the Conservation Program of the Puerto Rican Parrot by the U.S. Fish and Wildlife Service, shows that very few polymorphic sites exist in the genomes of the surviving birds (none of which are located within coding regions), consistent with the low genetic diversity found across the nuclear genome of this species [25,83]. While it has been noted that mtDNA variability in birds is significantly lower than in mammals, the estimate of nucleotide diversity [122] based on the few differences we have found in the full mitochondrial DNA of the Puerto Rican parrot is an order of magnitude lower compared to what is reported in the literature [123] and amounts to only 0.00042.

We found that *A. vittata* is not the only species with low diversity: the observed mitogenome diversity in *A. agilis* from Jamaica is also very low. While this observation is based on a small number of rescued birds, it is consistent with reports of declining numbers in this species and the vulnerable status assigned by the IUCN Red List [121]. The allopatric speciation scenarios (Figure 5 and Appendix A) imply that the two species currently found in Jamaica have been separated for at least two million years, and by the time *A. collaria* had arrived in Jamaica, the biological differences with the resident *A. collaria* species had already been established. This would explain why the two Jamaican amazons are so different: they are separated by almost two million years of separate evolution which would account for many genetic, morphological, and behavioral differences between them. While mitochondrial data are not suitable to explore genomic signatures of evolution, but the full genome sequences can eventually be searched for the chromosomal regions of divergence and to identify genes responsible for adaptation in each of the species [86].

Today, the two species occur sympatrically in significant numbers, particularly those in the mid-elevation wet limestone forests of the John Crow Mountains and Cockpit Country and have been observed to roost communally [124,125]. The green coloration of both species may be an adaptation against the avian predation. The plumage of *A. agilis* makes it blend well with the rainforest foliage: it is mostly emerald-green with small patches of red color on the wings, as well as flecks on the head. The plumage of *A. collaria,* which lives in the moist lowland, mountains, and mangrove forests, and often feeds on plantations and rural gardens, is also green, but with a pink throat and neck. Nest predation is the main factor limiting reproduction for most birds [126,127] and it is the major factor for the parrots in Jamaica, as the nesting success in either species is close to 50% [124,125], which is low compared to 60–70% for the other island *Amazona* parrots that nest in tree hollows [15]. *A. collaria* may require larger trees in which to successfully rear young, and the nest site availability may be a stronger limiting factor for them [128]. Agonistic interactions have been observed between the individuals of the two species on several occasions, with *A. collaria* dominating *A. agilis*, but the degree to which the larger *A. collaria* excludes the smaller *A. agilis* from the competitive nest sites remains unknown [124,125]. This, along with the feeding habits may be delineating the ecological niches of the two species today.

In addition to the species from the Greater Antilles, to provide an outgroup for the phylogenetic analysis, we have sequenced, assembled, and made available the mitogenome of the red-browed amazon (*Amazona rhodocorytha*) endemic to the Atlantic Forest in eastern Brazil. The *A. rhodocorytha* population is threatened both by habitat loss and by being captured for the trade of wild parrots and is classified as vulnerable by the IUCN [12,129]. There are still only a few studies on the genetic variability of Brazilian parrots of the genus *Amazona*, and we hope that this sequence will help conservation geneticists to further study this species.

## 5. Conclusions

With today’s third-generation sequencing technologies, such as Pacific Biosciences and Oxford Nanopore long-read sequencing [130,131], mitochondrial genomes are still much easier to assemble than complete nuclear genomes and carry a substantial amount of valuable information in terms of interspecific and intraspecific phylogenetic signal. Mitochondrial sequences have long been used in phylogenetic and population studies because of their mode of inheritance, which does not involve recombination [132]. Due to the predominantly maternal transmission of mitochondrial genetic material, it is possible to reconstruct mitochondrial phylogenetic trees and track ancestors of many diverged animal populations as they migrated from their geographical origin [133]. These advantages had made mitogenomes an important marker for evolutionary studies long before full nuclear genome sequences became available [133,134]. At the same time, mitogenome-based phylogenetic studies have serious limitations, mainly because an entire mitochondrial genome behaves as a single locus, and a mitochondrial phylogeny does not fully describe the evolutionary history of a sexually reproducing species, as half of the lineages are lost in each generation. Phenomena such as incomplete lineage sorting and gene flow following speciation events interfere with proper interpretation of phylogenetic trees built using mitochondrial genetic data alone [133]. Therefore, the results based on the mitogenomes of Greater Antillean *Amazona* parrots do not yet allow us to properly address questions about historical population sizes and the possibility of pre-and post-divergent gene flow between the extant species.

Examination of additional complete mitogenomes, as well as nuclear genomes of all Greater Antillean species and subspecies, should empower subsequent assessments of sequence variation, population diversity, inbreeding depression, demographic vital rates, and viability, and this is especially important in the case of the critically endangered Puerto Rican Amazon (*A. vittata*), one of the rarest parrot species still surviving in the wild [85]. Taking into consideration the significance of parrots to the ecology and history of all the Greater Antillean islands, we believe it is essential to understand how these species evolved and how they adapted to specific island environments. We look forward to the emergence of a larger set of whole-genome data and future analyses that will provide valuable insights into the evolution and functioning of the genomes of these unique bird species.

## Figures and Tables

**Figure 2 genes-12-00608-f002:**
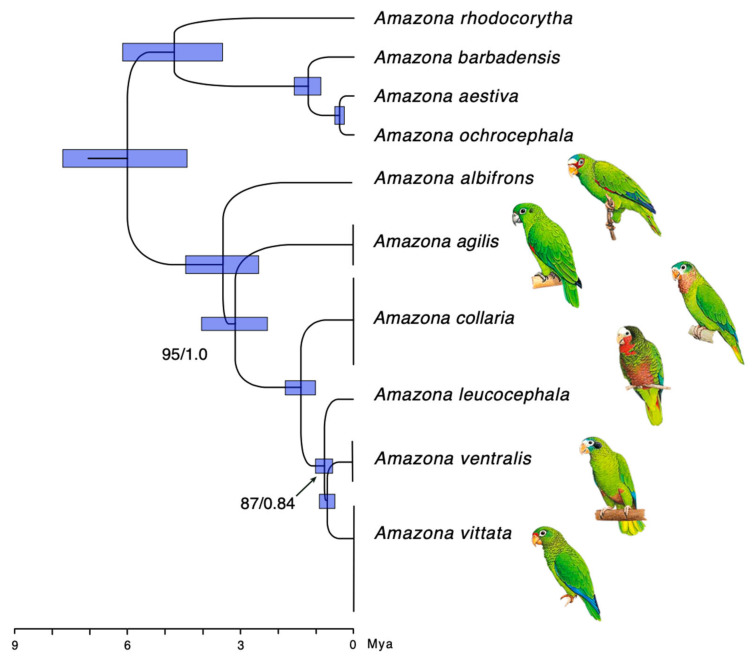
Fragment of the time tree for the Greater Antillean amazon parrots based on Bayesian inference in the larger context of Neotropical parrot evolution. The complete phylogeny is shown in Appendix A. All nodes except the ones indicated with numbers received both maximum likelihood bootstrap (100%, based on 1000 pseudo-replicates) and Bayesian posterior probability (1.0) support. The timescale is in millions of years ago (MYA). The tree was rooted with mitogenomic sequences of macaws and parakeets (all sources listed in Appendix A). The Bayesian analysis with a relaxed clock model rendered the same tree topology with nearly identical node ages and support values (see Appendix A). The maximum likelihood and maximum parsimony trees (Appendix A) are congruent with the tree shown.

**Figure 3 genes-12-00608-f003:**
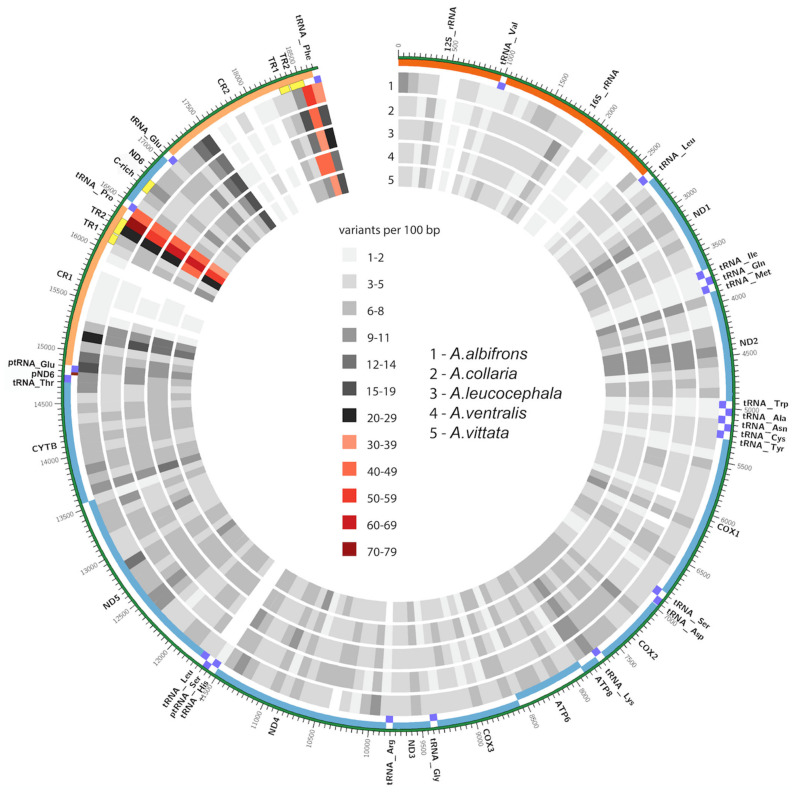
Mitogenome assemblies of the five species of *Amazona* from the Greater Antilles compared to the black-billed parrot (*Amazona agilis*) from Jamaica showing the density of variants. Consensus sequences were used if more than one individual was available for the species. The outer track shows the features from the mitogenome annotation. The diversity tracks (01–05) show variant density (per 100 bp) of the pairwise differences between *A. agilis* and (from the outside in): (1) white-fronted parrot *(A. albifrons*), yellow-billed parrot (*A. collaria*), Cuban Parrot (*A. leucocephala*), Hispaniolan Parrot (*A. ventralis*), and the Puerto Rican parrot (*A. vittata*). Pairwise absolute character differences among five species of Greater Antillean Amazona parrots based on alignment to *A. agilis* are given in Appendix A. The figure was generated using Circos [41].

**Figure 4 genes-12-00608-f004:**
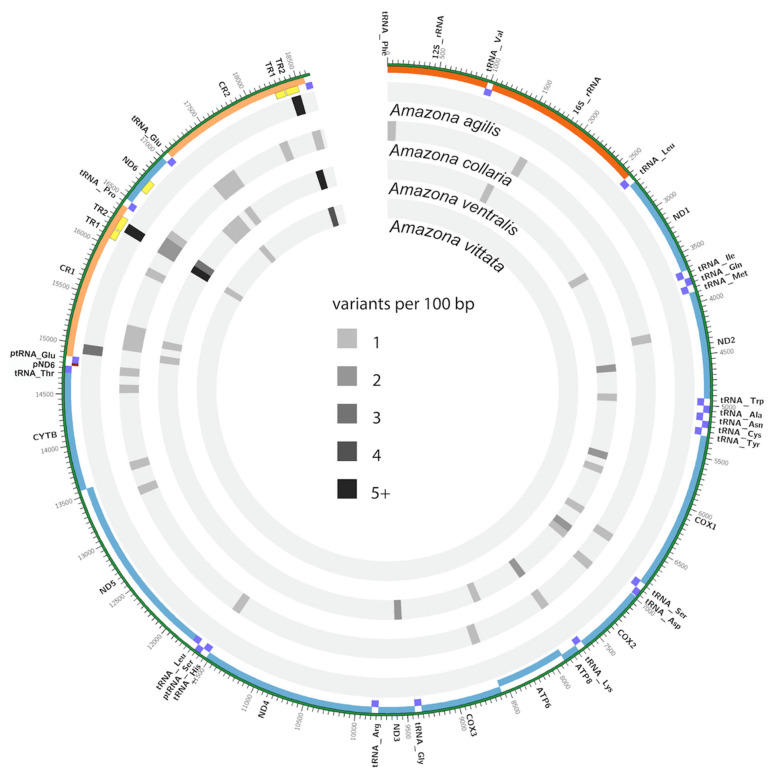
Circos plot showing intraspecific variation in four species of *Amazona* parrots from the Greater Antilles. Despite having the most individual sequences, *A. vittata* shows very little variation compared to other parrot species. Each track shows the density of variation (variants per 100 bp) between all the individuals sequenced from a single species. *A. vittata* n = 10; *A. ventralis* n = 2; *A. agilis* n = 2; *A. collaria* n = 7. The figure was generated using Circos [27]. Files containing locations of all polymorphisms are presented in Appendix D. Absolute character differences within four species of Greater Antillean are given in Appendix A. Visualizations of the alignments can be found in Appendix A.

**Figure 5 genes-12-00608-f005:**
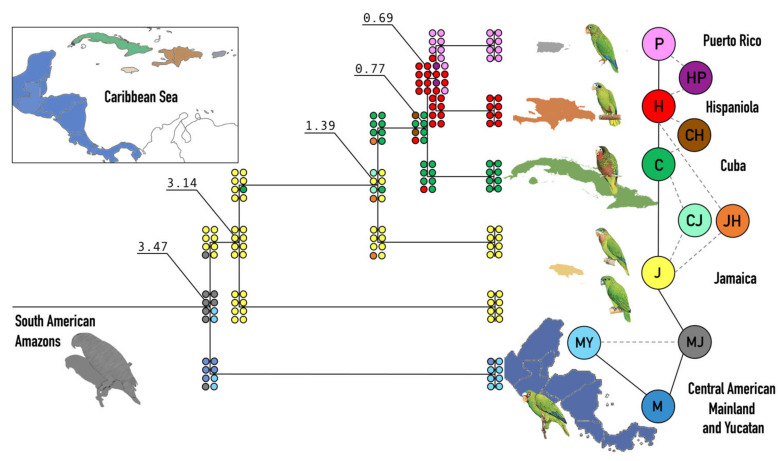
Visual summary combining speciation patterns predicted by eight different biogeographical models tested to infer possible speciation and dispersal routes. The eight models (DIVALIKE + X, DIVALIKE + J, DIVALIKE2 + X, DIVALIKE2 + J, DEC + x, DEC + J, BAYAREALIKE + X, BAYAREALIKE + J) from the *BioGeoBEARS* package [54] using a distance matrix based on the shortest distances between the islands scenario and evolutionary history of the Amazon parrot species in the Greater Antilles based on the data from this study (Supplementary Figure 2 and Appendix A). All eight models are fully described in Appendix A and represented by eight circles at each node (following the order in Appendix A) and summarized in Table 1. The colors of the circles represent the area where the speciation was predicted to occur by a specific model (M—Central American mainland; MY—Central American Mainland and Yucatan; MJ—Central American Mainland and Jamaica; J—Jamaica, CJ—Cuba and Jamaica; JH—Jamaica and Hispaniola; C—Cuba; CH—Cuba and Hispaniola; H—Hispaniola; HP—Hispaniola and Puerto Rico; P—Puerto Rico). Numbers on the arrows show approximate ages of the internal nodes in millions of years listed in Appendix A.

**Table 1 genes-12-00608-t001:** Summary of the major events predicted by different biogeographical models tested to infer possible speciation and dispersal routes using the *BioGeoBEARS* package [67] using a distance matrix based on the shortest distances between the islands (Appendix A). Most of the models (6 out of 8) agreed that there was an original MJ gene pool connecting mainland (M) and Jamaica (J). The existence of other shared genetic pools is supported by one or two models (see also Appendix A for each model’s illustration).

	Model	DIVALIKE (A)	DIVALIKE + J (B)	DIVALIKE2 (C)	DIVALIKE + J2(D)	DEC (E)	DEC + J (F)	BAYAREALIKE (G)	BAYAREALIKE + J (H)
	Model ID	A	B	C	D	E	F	G	H
	***Feature***								
1	Common MJ gene pool	*+*	*+*	*+*	*+*	*+*	*−*	*+*	*−*
2	Common MY gene pool	*−*	*−*	*−*	*−*	*−*	*+*	*−*	*+*
3	Sympatric Speciation in J	*−*	*+*	*+*	*+*	*−*	*−*	*−*	*+*
4	Common CJ gene pool	*+*	*−*	*−*	*−*	*+*	*−*	*−*	*−*
5	Speciation after J to C dispersal	*−*	*−*	*−*	*−*	*−*	*+*	*−*	*−*
6	Return dispersal from C to J	*−*	*−*	*−*	*−*	*−*	*+*	*−*	*−*
7	Stepping stone dispersal from C to H to P	*+*	*+*	*+*	*+*	*−*	*+*	*−*	*+*
8	Common CH gene pool	*−*	*−*	*−*	*−*	*+*	*−*	*−*	*−*
9	Common JH gene pool	*−*	*−*	*−*	*−*	*−*	*−*	*+*	*−*

## Data Availability

All new data generated for this study have been uploaded to NCBI: PRJNA496322.

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
