# Peer review of "Molecular Phylogeny and Evolution of Amazon Parrots in the Greater Antilles"

_genes, 2021, doi:10.3390/genes12040608_

Round 1

Reviewer 1 Report

This manuscript sequences complete mitogenomes from five species of Amazona parrots, and outgroups, to examine relationships among species, reconstruct patterns of colonization, and explore levels of genetic diversity. The methods are appropriate, and the results include a well-resolved phylogeny. The biogeographic analyses indicate several possible scenarios regarding patterns of colonization. Although sampling was limited within species (1-10/species), genetic variation appeared limited, as might be expected for limited island populations that are of conservation concern.

General Comments:

While there is a lot of interesting information in this study, sometimes the goals of the manuscript were hard to determine. I think shortening and increasing the focus would make the key results and their interpretation more accessible to many readers, and thus increase its visibility and interest.

In places, the manuscript feels almost more like a methods manuscript on the challenges of assembling mitogenomes from different data types, or a manuscript about mitochondrial evolution (e.g., the first page of results and first part of discussion are focused on these issues - there are three images of the mitochondrion, and only two on evolutionary patterns). Some of this, although potentially useful (and likely represents a lot of work), is not directly related to the key goals of the manuscript and I felt detracted from the goals. Though the details should still be included in methods, as it is important to clarify what was done, some of the results about this could be moved to the supplementary material to keep the focus more on parrot biology/evolution rather than methodological issues.

Overall, the writing could use some improvement, and my comments are largely addressing structure, repetition, and in a few places, providing more details - particularly for methods, where some approaches have insufficient methods to follow what was done. In general, the results repeated many details of methods, and the discussion included many details of the results, and this could be reduced to make the manuscript shorter and more clearly focused on the key issues. In some places, some reorganization might be helpful, including making sure tables and figures (both main text and supplementary) are presented in order they are referred to in the manuscript, and that they are referred to correctly.

The discussion could use some broadening and greater inclusion of literature for other taxa (other bird species, other island endemics, etc.) so that these results are clearly placed into the broader context of what we know.

Specific Comments:

Lines 52-62 (most of first paragraph): This discusses islands as important places for research, yet this manuscript does not study population on islands, but rather speciation patterns. I might focus more on information that leads to the questions being studied.

Line 63-64: But how many endemics - that would seem to be the more relevant measure.

Lines 84-93: Consider moving this paragraph later in the introduction (after giving dispersal scenarios?).

Line 101: Given that reconstructing the colonization history is a key objective (based on what is in introduction), why is Figure S1 hidden in supplementary? I would suggest this is more important to understanding the goals of the paper than either Figures 1 or 3, both of which I feel could be moved to supplementary material.

Lines 105-108: I am not clear on relevance of blue primary coverts as you mention red forehead (lines 102-105) then come back to red forehead (line 110). The blue primary coverts is not clearly linked with the colonization scenarios.

Lines 121-123: Given that even complete mitogenomes likely represent a single locus, why were NGS methods that target nuclear regions not used?

Line 144: How many individuals of A. agilis and A. collaria? It is not clear as written (this is in supplementary, but numbers are given for other species so this should be done consistently).

Lines 148-150: Was there any verification that pet birds were correctly identified and had not experienced any hybridization in their past (often a problem for taxa maintained in captivity for long periods)?

I found section 2.1 (assembly and sequencing) hard to follow in terms of what method was used on what species.

Based on Table S1, it sound like most of these first few paragraphs is specific to A. vittata. Perhaps it should be separated as to what you did for that species, and then what was done for other species? For example, rather than saying you used multitple strategies, begin with "For A. vittata, we used multiple strategies….". Then later on you could begin with "For other species, we used X strategies….".

Also, until I got to section 2.2, I had not realized you used a different method for the samples listed in Table S1 than what was given. This all needs to be written more clearly as I had to look carefully at both supplementary files and the main text to feel I understood what was done.

Lines 161-163: Amplification of short pieces can lead to amplification of numts (nuclear copies of mitochondrial regions) - particularly since the more slowly evolving numts often conserve priming sites better than the actual mitochondrial copies. What efforts were made to ensure that the final assemblies of the Sanger data were actually mitochondrial and did not potentially include numts? This is addressed later, so maybe it is not necessary to mention here. But if you began by stating you used a mixture of Sanger, Illumina and PacBio, it would clarify initially that this was not an issue.

Lines 179-180: For readers not familiar with this PacBio version, can you indicate the read length (or range of lengths) obtained? It is on the figure (or could be extrapolated from that), but that requires looking carefully at the figure.

Lines 204-217: Can you briefly indicate coverage of the mitogenome with this data (the statistics in Table S3 are for the entire genome, not mitogenome so it is hard to assess quality of the mitogenome you might have assembled)?

Any efforts to ensure you had mitochondrial, not numt, reads in these assemblies? - I realize this is addressed later, and that may be sufficient.

Lines 216-217: I am not sure what “best practices” are for extracting complete mitogenomes from various types of NGS data. Can you please briefly summarize, since you provide no citation? This may clarify the numt issue I mentioned in the previous comment.

Lines 225-230: Some methods for hybridization conditions, washes, etc. should be provided. Presumably samples were bar-coded?

Section 2.3 is titled "mitogenome assemblies and annotation" and 2.1 was titled "mitogenome sequencing and assembly". Both section2.1 and 2.2 described some assembly. Consider reorganizing to make all assembly information grouped together consistently (or put assembly with each species/method used, and if so, retitle sections to reflect that organization).

Talbe S1 indicates the use of IonTorrent, yet I do not see that reflected in either sequencing or assembly sections.

Line 271 (and elsewhere in methods): How much manual curation was required and what was the nature of the types of curation that were necessary? Please clarify whenever you did manual curation what was done.

Line 291: Appendix 3 links to a google doc. I think it would be preferable to put this into a permanent archive, such as Dryad, FigShare, Zenodo (or as supplementary material, if not too large).

Section 2.4: This section includes alignment, samples included, and phylogenetic analyses. I might retitle to Alignment and Phylogenetic analyses.

Lines 313-314: Some studies have aligned and analyzed the CR across avian families (or even within an order) and found results congruent with other parts of the mitogenome. So it could be quite useful, particularly for close relationships - and out side of some of the core region, may not be under much selection (so could be better modeled than the more challenging to model PCGs). It can be more challenging to align, and is likely not assembled as well, so there could be arguments against using it.

Line 314 -"mapping bias of indels: I have not seen this noted in other studies - is there a citation that this can be a problem for non-coding regions?

Line 319: What were alignments inspected for?

Lines 320: Within PCGs, were 1st, 2nd, and 3rd position considered as separate paratitions since they both vary in rate but also other parameters like GC content? Based on the end of line 322, it sounds as if you did not partition among PCGs, but lumped them together, as well as combining both rRNAs and all tRNAs into a partition (so also did not separate out by codon position). I might write that more clearly, as most mitogenome papers separate these categories for PartitionFinder analysis, so many may assume you had separated these.

Line 337: why was a strict clock model assumed?

Lines 356-362: You describe using MP, after you describe clock analyses.

However, in lines 306-307, only ML and Bayesian are mentioned as methods used for phylogenetic estimation - not MP. I would edit lines 306-307 to include MP. I would also recommend moving the MP details earlier with other phylogenetic tree estimation method.

I also am not clear why you used a single mitogenomes per species, and only one outgroup for MP analyses. If you feel MP adds something important, then I would make it comparable to the other analyses that you conducted.

Biogeographic analyses: Although there are a lot of details of how BioGeoBears works, it would be helpful how you assigned the geographic areas to the taxa in your tree (given you have many mainland taxa as well). Please provide more details of how you used the program.

Lines 408-418: Given that the majority of data is NGS and that most new mitogenomes are sequenced using NGS method, I wonder whether including a lot of details on the Sanger results is the most important result for the main text (versus supplementary material).

Lines 408-433: I also wonder whether some of this might be better in methods, as it really explains why you did various methods, but does not address the key goals laid out in the introduction (phylogenetic hypotheses, patterns of speciation, and genetic diversity).

Line 445: You refer to figure S4 first in the manuscript, then S5 and S6 before getting to S2, S3. I would renumber figures (in text and in supplementary) to reflect when they are first referred to. It would make it easier for a reader to follow what was being done, and not worry that something had been skipped.

Line 482: Can you define what you mean by congruent? Mostly the same? Identical?

Lines 483-488: If MP/ML/Bayesian are identical, then I think you could shorten this and not describe those relationships again.

Lines 503-511. Much of this paragraph was included in the methods, and I do not think it needs to be repeated here.

Line 510: Figure 3 is an image of mitogenomes - not biogeographic results. Should this be figure 5? I found figure 5 useful, but do not see it referenced in the text.

Line 556: I might refer to table S5 in methods, rather than here, since it details the model parameters used in analyses.

Figure 3, and related results: Why was A. agilis chosen as the mitogenome for comparison? It feels this should be justified. Given that you obtained 3 mitogenomes for this species, which one was used?

For species with multiple individuals, how much of the results in Figure 3 reflect intraspecific variation versus interspecific variation?

While I understood what Figure 4 was conveying, I did not understand what was learned from Figure 3 that was relevant to the questions being addressed. This should either be clarified (its importance justified), or this figure (and associated text) might be excluded (or moved to supplementary material).

Lines 582-590: This mostly repeats methods and could be deleted.

Line 594: Given that you had between 2 and 10 individuals, were results adjusted to reflect the differing sampling strategies? Obviously more variation might be observed if you sample 10 than 2.

Line 622: Should this be figure 2, and not figures 1 and 3?

Discussion, first paragraph: That the mitogenomes were similar to other closely related parrots does not seem to be a key finding, and is not one of the stated goals. I might rewrite this paragraph to give an overview of the key findings relative to the questions posed in the introduction, and then elaborate on other points later in the discussion.

Lines 657-695: Much of this material is not about phylogeny, but instead about challenges of getting data, estimating clocks, etc. I would move to a separate section. I also think much of this could be shortened (e.g., the challenges of de novo assembly are not directly relevant) - though it is critical to clarify this represents a single gene history.

I would keep this subheading (Phylogeny) to put your phylogenetic (and clock?) results into the broader literature about parrots and other taxa in this section.

In the discussion, some of the clock results are discussed under the phylogeny section, some with biogeography. I wonder if it would be better to include them together into one section (potentially with biogeography, so those results could be put into a time framework).

A lot of the biogeography section goes over the model results. Information that links the results to what is known in other taxa, other information that might help resolve differences among models, etc. are included, but hard to find. I would shorten this, and focus less on the models, but more on putting the results into context - are some models more consistent with other information or other species that might suggest they are more likely?

Lines 713-726: This was also detailed in the introduction. Here I would focus on the results obtained and which hypothesis those supported, etc.

Table 1: Please make the + and - signs much larger so they are easy to see on a smaller screen.

Lines 869-872: The maternal inheritance traces female patterns, not male. So you get part of the story, not the entire picture.

Lines 889: Diversity should also link to sample sites - were birds likely caught from a single population? (or maybe there is only a single panmictic population on many of these islands?). If you sample a single population, but there are others, then you underestimate species diversity.

How does diversity of these parrots compare to other taxa - other island (or endangered) birds?

Reviewer 2 Report

The new revised version of the MS “Molecular phylogeny and evolution of Amazon parrots in the Greater Antilles” has been improved and I have no major methodological issues. However, additional editing can further improve the MS considerably. I think most sections can be shortened and streamlined. The authors should focus on the main message and the most significant results. Below I give a just few examples.

Repetitive parts

There are still several repetitive parts in the text and one example is found already in the abstract, where line 34-36 “Information from the assembled and annotated mitogenome maps provides information on sequence variation, population diversity, and can help design future population and conservation studies” and line 45-47 “This analysis contributes to understating history of speciation, empowers subsequent assessments of sequence variation and helps future conservation efforts” give more or less the same information.

Redundancy

Especial the results and material and methods sections include a lot of explanatory text. Some of this is obviously important and necessary but in my opinion the authors sometimes explain models and programs in such detail that the text become difficult to follow, such information can be transferred to supplementary material (or just refer to the original publications). Especially I found the section “Dispersal and speciation” unnecessary detailed. For example, why not for example replace the section “Biogeography has a number of methods for inferring ancestral geographic ranges on phylogenies, many of which use different, often conflicting assumptions. Probabilistic modeling of geographic range evolution using the R package BioGeoBEARS [66,67] provides a statistical platform for evaluating multiple biogeographical models given the observed data from a phylogenetic analysis combined with geographical data, such as sizes and distances between landmasses. It uses standard statistical model choice procedures, as long as multiple models are available” with something like “We used the R package BioGeoBEARS to infer the biogeographical history as multiple biogeographical models can be tested”. I also wonder if it is logical to test all models or can some models a priori be excluded, as they may be unrealistic?

The authors also need to check the grammar and spelling (see for example the first sentence in the discussion).

Author Response

This manuscript is a resubmission of an earlier submission. The following is a list of the peer review reports and author responses from that submission.

Round 1

Reviewer 1 Report

The reviewed article analyses biogeographic and phylogenetic aspects of selected Amazon parrots from Grater Antilles. Overall, the manuscript subject is very interesting and current, and as such may be interesting to a wide scientific audience. In particular, authors produced new data (22 samples from seven species were sequenced for mtDNA) to evaluate phylogeny of analysed group of parrots, evaluate proposed speciation hypotheses and to check the intraspecies genetic diversity for some researched species (with more than one individual). Unfortunately, in my opinion, this version of manuscript needs to be improved to accomplish requirements of publishing it in Genes.

During the review, I found several problems, I would like to address to:

  • The speciation of the Amazona genus is a very interesting topic, especially in terms of determining the direction of colonization. As authors described in introduction there were several different scenarios, which tried to explain this process. However, in the case of Island species, all presented scenarios concerned only islands colonization from the mainland. In this context, I would like to know why authors, did not mentioned and discussed results presented by Urantowka et al., (2014) who showed that descendants of the Yellow-headed Amazona group colonized Central America through the Isthmus of Panama as well as southward regions of South America from northernmost regions of South America. Phylogenetic position of two Lesser Antillean Amazons ( arausiaca and A. versicolor) as well as Amazona barbdensis (island/mainland habitat) turned out to be crucial for the final conclusion about the “unusual” direction of land colonization from the islands. Moreover, phylogenetic position of two additional Lesser Antillean taxa (A. imperialis and A. guildingii) as well as Greater Antillean A. agilis raises the question about original direction of Amazona taxa colonization (Silva et al., 2017; Figure 12). The aforementioned analyses were performed with the use of only a few short mitochondrial sequences but in my opinion, authors should take them into account when discussing results. Especially, that analyses cited in introduction and discussion section were based on similar or even identical molecular markers.
  • As mutually exclusive scenarios are currently suggested in the literature, It is obvious that the use of full mitochondrial genomes of Amazona taxa is necessary to get a definitive answer to the question about the original direction of colonization. Therefore, mitochondrial genomes of Greater Antillean Amazons ( agilis, A. collaria, A. leucocephala, A. ventralis, A. vittata); Central American species (A. albifrons) and South American taxon (A. rhodocorytha) may (and in my opinion will) occur particularly important role in phylogeographical analysis. I do not understand, why authors did not amplify and sequence mitogenomes of some other Amazona taxa. The use of taxa representative for each known Amazona clade and important from the view of their geographic distribution is really essential to obtain final answer. Moreover, it was shown that the number of used taxa may have a great influence on the topology of the obtained phylogenetic trees (Tamashiro et al., 2019). I really encourage the authors to use mitochondrial genomes of as many Amazona taxa as it is possible to propose the final colonization scenario. Furthermore, Amazona xantholora taxa has to be used in the analysis to avoid making Amazona albifrons an arbitrary outgroup. Especially that the phylogenetic position of A. agilis was shown to be dependent on the used tests (Silva et al., 2017; Figure 13). I really appreciate the contribution of the authors of this manuscript to obtain 22 mitochondrial genomes of seven Amazona taxa and preparing the presented results. Unfortunately, in my opinion, the dataset which was used is insufficient to draw the conclusions presented in this manuscript. I also hope, that the presence of mitogenome of Lesser Antillean species (Amazona guildingii) in the GenBank database (MN356210.1) will occur helpful for the authors to continue their analyses.
  • Moreover, the authors of the article did not provide information on the subspecies affiliation in the case of leucocephala (two Cuban subspecies) and Amazona albifrons (three known subspecies). Of course, only one subspecies of Amazona albifrons inhabit Yucatan peninsula but I really do not know if albifrons nana sample was used based on the information:

“The white-fronted amazon (A. albifrons, Yucatan peninsula, Mexico), and the red-browed amazon (A. rhodocorytha, Brazil) samples consisted of plucked feathers mailed by the owners of pet birds in private collections to the University of Puerto Rico at the personal request of the aviculturists from the USFWS Puerto Rican Parrot Recovery Program.”

  • The authors of the manuscript excluded control regions from the analyses, as they considered them to be too variable for the interspecific scale. I am curious what was the background of this decision as I did not found any analyses done to prove this statement nor any citation. Furthermore, results showed for other Arinae tribe, proved that control regions can be really important to obtain true phylogenetic relations even between taxa belonging to different genera (Urantowka et al., 2017). So, I strongly suggest to add control regions to analysis (at least their central conservative regions excluding the microsatellite sections).
  • I also hope that authors of the work will meet the requirements for public availability of the published data, as in the presented manuscript I did not find any information about GenBank numbers and even the length of finally obtained sequences. The lack of such information is unacceptable.

To conclude, in my opinion, the manuscript in a present form cannot be accepted. Having not enough data, the authors of the manuscript  did not provide compelling evidence that the species of the Greater Antilles colonized the islands from the Yucatan Peninsula. Rather, it was an assumption based on articles mentioned in the Introduction section. It could be consequence of the fact, that the authors of the study did not refer to the latest literature in which the phylogenetics of the genus Amazona were also presented (Urantówka et al., 2014; Silva et al., 2017).  Actually,  obtained sequences of mitochondrial genomes can be published only in the context of Intra- and interspecific genetic differences, which is really interesting and widely discussed in the manuscript. Consequently, the present manuscript should be rejected. However, some other manuscript variants (depending on the Authors` decision) should be prepared for review. The manuscript concerning Intra- and interspecific genetic differences between so far obtained mitogenomes or new manuscript concerning colonization scenario but based on broader set of mitogenomes. I really hope to see such variants published soon.

Reviewer 2 Report

The manuscript “Molecular phylogeny and evolution of Amazon parrots in the Greater Antilles” use mitochondrial genomes to study relationships of Amazon parrots, how they colonized the islands of the Greater Antilles, and to assess genetic diversity. Most downstream analyses appears sound, but there are issues that the authors needs to address.

The authors have used multiple methods to recover mt genomes and some of these are logical. However, others seems just to be byproducts from other projects. If that is the case I think that should be stated, e.g. what is the logics of using 10X Chromium to construct a mt genome?  

The Amazona vittata mt genome has been assembled using multiple types sequence data, including long-read sequencing (PacBio). However, the extreme Illumina coverage of the control regions and almost lack of sequencing mapping in the intermediate region (Fig 1) suggest that the genome may suffer from assembly problems. The authors has noted it and state that an access of NUMTs mapped to the region might explain this observation.  However, given that the authors state that the similarity between NUMTS and mt regions ranges between 64.8% and 95.2% it should be possible to detect NUMTs and remove them. The authors also need to explore and explain why there seems to be very few reads that have mapped to intermediate region (none is some regions?). Without a more proper examination of these issues, I would not trust this part of the mt genome.

Several of the mt genomes have been recovered by mapping against the newly constructed A vittata mt genome, that is an ingroup in the study. To ovoid that mapping bias influence the results in phylogenetic studies, one normally use a taxon that is equally distant to the focal study group, as reference for mapping. As this study is based on the relative conserved mt genome it might not be an issue. However, potential for example insertions might have been missed and I thus think the authors should address this potential problem.

I found the MS in its current form is rather difficult to follow and that the text can be revised and condensed . There are quite a lot of similar text sections in e.g., methods and the results and in introduction and discussion. The headings could be better adjusted, for example starts the “Phylogeny” section with a discussion of mitogenomic diversity.